# Cesarean delivery surgical techniques in Africa: A survey study from Ethiopia

**Wondimu Gudu**[1], **Zekarias Taye Sisay**[1], **Mekitie Wondafrash**[2], **Abraham Fessehaye Sium**[1]*

**1** Department of obstetrics and Gynecology, St Paul's Hospital Millennium Medical College, Addis Ababa, Ethiopia, **2** St. Paul Institute for Reproductive Health and Rights, Addis Ababa, Ethiopia

* abrahamfessehaye4@gmail.com

## Abstract

### Objective

To describe the surgical techniques of Caesarean delivery (CD) practiced by Ethiopian Obstetricians and Gynecologists.

### Methods

A descriptive survey study was conducted in Ethiopia from March 1, 2021 to April 30, 2021. Members of the Ethiopian Society of Obstetrician and Gynecologists were randomly selected and their Cesarean delivery surgical techniques were explored. Data were analyzed using IBM SPSS statistics 22. Simple descriptive analysis were employed and frequencies and percentage were calculated to present the data.

### Results

A total of 258 obstetricians and Gynecologists practicing in Ethiopia were approached with a response rate of 97.3% (251/258). Double layer closure of uterine incision (98.4%) and subcuticular closure of skin wound (96.4%) are practiced by most of the participants. There was a large difference in practice of blunt versus sharp fascia extension (43.3 vs 55.8%), cephalo-caudad versus lateral uterine incision extension (58 vs.39%), and closure versus non-closure of pelvic and parietal peritoneum (57.4 vs 42.6, and 39.8 versus 60.2%).

### Conclusions

Blunt and sharp fascia extension, cephalo-caudad and lateral uterine incision extension, closure and non-closure of the pelvic and parietal peritoneum are practiced by similar numbers of Ethiopian Obstetricians and Gynecologists. This demonstrates a wide variation exists in the techniques of Cesarean Delivery across Ethiopia.

**Data Availability Statement:** There are restrictions on publicly sharing the complete minimal data for this study because it contains potentially identifying patient information and it is restricted to maintain patient confidentiality. Data are available upon

request from Birhan Wale (Department of
Obstetrics and Gynecology, St. Paul's Hospital
Millennium Medical College (SPHMMC)) via email
(birhan.wale@sphmmc.edu.et) for researchers
who meet the criteria for access to confidential
data.

**Funding:** The author(s) received no specific
funding for this work.

**Competing interests:** The authors have declared
that no competing interests exist.

## Introduction

Caesarean delivery (CD) is one of the most common surgical procedures in the world which is
utilized as a safe option of delivery through a women's abdomen, when vaginal birth is consid-
ered dangerous to the mother or baby [1]. The rise in CD rates globally in recent years has
prompted increased interest in the indications, complications, and techniques involved with
this procedure [2].

Despite existence of strong recommendations from American College of Obstetricians and
Gynaecologists (ACOG) [3] and the National Institute of Health and Excellence (NICE) [4],
there are variations in the caesarean section techniques practiced across countries and even
within one country [5]. A 2016 US survey study among 247 members of ACOG found that
similar numbers of obstetricians either reapproximate or leave open the rectus muscles and
parietal peritoneum at CD, suggesting that wide variation in practice exists [6]. Another simi-
lar study from UK, which aimed to determine the techniques of CD among members and fel-
lows of the Royal College of Obstetricians and Gynecologists (RCOG), found that a wide
variation exists in the surgical techniques practiced by the professionals [7].

Although mutiple studies have documented the variation in the techniques of CD utilized
by obstetricians in high-income countries, there is no data on this from low-income coun-
tries, including Sub-Saharan Africa. Standarizing procedure technique and having a uni-
form practice guide among different physicians and hospitals is essential as it avoids
confusion. Hence, it is imperative to explore what techniques of CD are practiced in such
low-income settings. According to arecent reports, the rate of cesarean section in Ethiopia
increased from 5.1% in 1995 to 16% in 2019 in an urban area and from 0.001 to 3% in the
rural setting during the same period, with the overall increment of CS rate across the coun-
try being 0.7% in 1995 to 2019 at 6% [8]. Among the Sub-Saharan Africa, Ethiopia-the sec-
ond largest population in Africa represents a wide scale of obsteric care practice including
CD with a large number of obstetricians in practice [9], though some of this burden is han-
dled by emergency surgical officers who provide emergency obsteric care including CD at
distric hospitals. The country is among the early champions in reproductive health with low
maternal mortality ratio (267/1000 live births), according to the most recent report [10].
This study aimed to determine the surgical techniques for Cesearan delivery used by Obster-
icians in Ethiopia.

## Methods and materials

This was a descriptive survey study conducted in Ethiopia over two months, from March 1,
2021 to April 30, 2021. The Ethiopian society of obstetricians and Gynecologists (ESOG) is
one of the biggest professional societies in Africa. It has more than 640 members. The CS rate
in Ethiopia is below the WHO recommended level, which is around 6% according to evidence
from 2000–2019 Ethiopia demographic and health survey data.

Among the members of the Ethiopian Society of Obstetricians and Gynecologists practicing
in Ethiopia, we randomly selected 258 participants. The inclusion criteria were: a registered
Obstetrician in Ethiopia and currently practicing CD, member of ESOG, working at private or
government institutions, practicing at teaching and non-teaching hospitals in different regions
of Ethiopia, at least one year of work experience as a specialist obstetrician, Obstetrics and
Gynecology residency training attended in Ethiopia, and willing to participate in the study.
The exclusion criteria were: incomplete response, incomplete data, emergency surgery officers
who practice CD, and resident obstetrics and gynecology physicians who were in training.

We prospectively collected data on socio-demographic characteristics, pre-operative CD
preparation (including prophylactic antibiotics and skin preparation methods), and CD

surgical techniques using an online Google Form web-based questionnaire as well as a printed version for some of participants who preferred to respond offline (through a postal mail). Participants were informed on the purpose and importance of the study and they were reassured about the confidentiality of the data that they were going to provide. The returned survey forms from the participants were checked for data completeness and consistency by the principal investigator. Ethical clearance was obtained from the institutional review board (IRB) of St. Paul's Hospital Millennium Medical College (Addis Ababa, Ethiopia) before data collection was commenced. Written informed consent was obtained from all study participants. SPSS version 22 was used for data analysis. Simple descriptive statistical analysis were employed to analyze the data. Frequencies and percentage were used to present the results.

## Results

### Sociodemographic characteristics

From 258 obstetrician and gynecologists who volunteered to participate in the study, 251 of them completed the survey questionnaire to result in a complete response rate of 97%. The mean age and years of experience of the participants were 36.7 years and 5.5 years, respectively (Table 1). Majority of the participants (96%) were general Obstetricians and Gynecologists. More than a third (39%) of them were practicing in the capital, Addis Ababa while the rest practiced outside the capital, in the other regions of Ethiopia. Almost equal numbers of participants were from government teaching and non-teaching hospital (38.6 vs.35.5%) with the rest 25.9% of them being from private practice (hospitals/MCH centers). Around 70% the participants had 1-5years (70.1%), followed by another 16.3% who had 6–10 years of experience. Only 12.4% of them had experience beyond 10 years.

**Table 1. Sociodemographic characteristics the study participants.**

| Variable | | Number | Percent |
|---|---|---|---|
| Age (Years) | Mean | 37.4 | |
| Current specialty/subspecialty | Generalist Obstetrician and gynecologist | 241 | 96 |
| | Maternal-fetal Medicine specialist | 4 | 1.6 |
| | Reproductive endocrinology and infertility specialist | 3 | 1.2 |
| | Gynecology oncologist | 3 | 1.2 |
| Place of practice | Teaching governmental hospital | 97 | 38.6 |
| | Non-teaching governmental hospital | 89 | 35.5 |
| | Private practice | 65 | 25.9 |
| Address of place of work | Addis Ababa city administration | 98 | 39 |
| | Oromia region | 58 | 23 |
| | Amhara region | 32 | 12.7 |
| | SNNPR | 24 | 9.6 |
| | Harari region | 12 | 4.8 |
| | Sidama region | 12 | 4.8 |
| | Other regions | 15 | 6.1 |
| Work Experience in years | Mean | 5.5 | |
| | 1–5 years | 178 | 71.1 |
| | 6–10 years | 41 | 16.4 |
| | >10 years | 32 | 12.5 |

## Cesarean delivery(CD) surgical techniques among obstetrician and gynecologists practicing in Ethiopia

The most commonly practiced antiseptics for skin preparation by the obstetricians was alcohol plus iodine (94.2% of them practiced this technique). Povidone iodine and iodine with Savlone were reported to have been utilized by3.4% and 2.4% of the participants (Table 2). Ampicillin and Ceftriaxone were the most commonly preferred prophylactic antibiotics for CD with 51.15 and 47.1% of the participants reported to have utilized them respectively. Around 22% of

**Table 2. Intra-operative techniques among obstetrician and gynecologists practicing in Ethiopia.**

| Variable | Category | Number | Percent |
|---|---|---|---|
| Skin preparation | Alcohol + Iodine | 236 | 94.2 |
| | Povidone + Iodine | 9 | 3.4 |
| | Iodine + Savlone | 6 | 2.4 |
| Prophylactic antibiotics | Ampicillin | 128 | 51.1 |
| | Ceftriaxone | 118 | 47.1 |
| | Other | 5 | 1.6 |
| Skin incision for CD | Pfannenstiel | 238 | 94.8% |
| | Joel-Cohen | 11 | 4.4% |
| | Others | 2 | 0.8 |
| Sub-cutaneous tissue dissection | Blunt | 146 | 58.2 |
| | Sharp with blade | 52 | 20.7 |
| | Sharp with scissors | 47 | 18.7 |
| | Mixed – blunt + Sharp | 6 | 2.4 |
| Fascia dissection | Sharp | 141 | 56 |
| | Blunt | 110 | 44 |
| Uterine incision in anterior placenta previa | LUST | 201 | 80.1 |
| | Low vertical | 27 | 10.8 |
| | Classical | 23 | 8.8 |
| Bladder flap creation | | 64 | 42.2 |
| Uterine incision extension | Cephalo-caudal | 146 | 58.2 |
| | Lateral extension | 98 | 39.1 |
| Placenta delivery | Mixed | 7 | 2.7 |
| | Continuous cord traction | 233 | 92.8 |
| | manually | 18 | 7.2 |
| Uterine incision in transverse lie | LUST | 125 | 49.8 |
| | Low vertical | 95 | 37.8 |
| | Classical | 31 | 11.2 |
| What technique of closure do you prefer for uterine wound? | Double layer | 247 | 98.4 |
| | Single layer | 4 | 1.6 |
| Stitching technique | Running locking | 28 | 11.2 |
| | Running non-locking | 222 | 88.4 |
| | Interrupted | 1 | 0.4 |
| Parietal peritoneum Closure? | Closure | 144 | 57.4 |
| | Non-closure | 107 | 42.6 |
| Visceral peritoneum Clouse? | Closure | 100 | 39.8 |
| | Non-closure | 151 | 60.2 |
| Uterine incision closure stitch | Polyglactin | 148 | 58.9 |
| | Catgut | 103 | 41.1 |

the participants reported that they practiced vaginal canal preparation before starting CD. The most commonly practiced skin incision during elective CD among the participants was Pfannenstiel (94.8%) followed by Joel-Cohen (4.4%). For skin incision for women with significant obesity or panniculus around 39% of the obstetricians used a higher incision above the panniculus, another 13.9% them used midline incision, only 7.2% of them incised below the panniculus whereas 27.4% them had any modification in the skin incision technique. Almost half of the obstetricians had not practiced separating the median raphe of the rectus abdominis muscle routinely (52.8%).

Most of the study participants (Ethiopian obstetricians) utilized extending the subcutaneous tissue bluntly with fingers (58.2%) followed by sharp dissection with the use of surgical blade and scissors (20.7 and 18.7%, respectively). Sharp extension of the fascia was used more than blunt extension (55.8% versus 43.8% of the participants). Among the obstetricians, 42.2% practiced bladder flap creation. The most commonly applied uterine incision extension was cephalo-caudad (both using left index finger and thumb for right hand individual, in 11.6%, and with index finger of both hands, in 46.6%) followed by lateral extension (with left index and right thumb fingers for right hand individuals, in 9.2%, and with index fingers of both hands, in 29.9%).

Around 80% of the participants used lower uterine segment incision for anterior placenta previa while 10.8% and 8.8% from the participants practiced low-vertical and classical incisions for the same indication (placenta previa) (Table 1). Continuous cord traction was applied to remove the placenta by around 93% of the participants. Around half from the participants utilized lower uterine segment incision for transverse lie. The remaining 37.8% and 11.2% of them practiced low-vertical and classical incision, respectively. The majority (98.4%) used double layer uterine incision closure and running non-locking stitching technique in 88.4%. The most commonly utilized stitch for uterine incision closure were polyglactin and catgut (58.9% and 41.1% respectively). There was a large difference in closure versus non-closure of pelvic and parietal peritoneum (57.4 vs 42.6, and 39.8 versus 60.2%) among the participants. The most commonly used skin wound closure technique was subcuticular (96.4%) with running non-locking suture (88.4%), running locking (11.2%) and interrupted suture technique (2.8%).

Analysis of surgical techniques based on place of practice demonstrated both bunt and sharp dissections were used to extend fascial incisions (36.4% vs 40%, 36.4% vs 35%, and 27% vs 25%, in governmental teaching hospitals, governmental Hospitals, and private practice, respectively) (Table 3). Routine cleaning of sub-cutaneous space with saline was practiced by 37.8%, 34.2%, and 28.1% obstetricians serving at government teaching hospitals, governmental Hospitals, and private practice, respectively.

**Table 3. Distribution of surgical techniques of CD among obstetricians in Ethiopia according to place of practice.**

| Techniques | | Place of practice | | |
| --- | --- | --- | --- | --- |
| | | Gov't teaching hospitals | Gov't hospitals | Private practice |
| Vaginal canal preparation | Yes | 32.7% | 32.7% | 34.5% |
| Thromboprophylaxis for CDs? | Yes | 51.4% | 25.0% | 23.6% |
| Extension of fascial incisions | Blunt | 36.4% | 36.4% | 27.3% |
| | Sharp | 40.0% | 35.0% | 25.0% |
| Routine creation of bladder flap? | Yes | 41.4% | 38.7% | 19.8% |
| Routine mopping of the uterine cavity | Yes | 39.4% | 34.5% | 26.1% |
| Subcutaneous space washing with saline | Yes | 37.8% | 34.2% | 28.1% |

## Discussion

Our study shows that there is a wide variation in surgical techniques for CD among Obstetricians practicing in Ethiopia. Blunt and sharp fascia extension, cephalo-caudad and lateral uterine incision extension, closure and non-closure of the pelvic and parietal peritoneum are practiced by these physicians. The commonly utilized stitches for uterine incision closure were polyglactin and catgut.

Recent literate indicates that wide variations exist in the surgical techniques used for caesarean section worldwide. Multiple studies also show that there is no difference among the different surgical techniques in terms of CD procedure outcomes. A randomized controlled trial, in which 15,935 women were recruited, found no differences in the primary outcome between: blunt versus sharp entry risk ratio 1·03 (95% CI 0.91–1.17), exterior versus intra-abdominal repair 0.96 (0.84–1.08), single-layer versus double-layer closure 0·96 (0.85–1.08), closure versus non-closure 1·06 (0·94–1·20), and chromic catgut versus polyglactin-910 0·90 (0.78–1.04) [11]. Another randomized controlled trial from UK, in which 3033 women were included, demonstrated there were no differences between the arms of the trial for the primary outcome: single versus double-layer closure of the uterine incision [relative risk (RR) = 1.00, 95% confidence interval (95% CI) = 0.85–1.18]; closure versus non-closure of the pelvic peritoneum (RR = 0.92, 95% CI = 0.78–1.08) [12]. A large Cochrane review (n = 17, 276, from 21 trails) has documented that nonclosure of parietal peritoneum was associated with a significant reduction of operative time and hospitalization [13]. Another similar review (based on analysis of 6 trials) that aimed to determine the impact of uterine incision types (sharp vs blunt extension) on maternal and perinatal outcomes found no statistically significant differences for febrile morbidity following sharp or blunt extension of the uterine incision, though blunt extension was associated with lower mean blood loss and need for blood transfusion [14].

Consistent with previous reports, our study found a wide variation in techniques of CD among obstetricians practicing in Ethiopia: blunt versus sharp fascia extension (43.3 vs 55.8%), closure versus non-closure of pelvic and parietal peritoneum (57.4 vs 42.6, and 39.8 vs 60.2%, respectively), Polyglactin versus catgut use for uterine incision closure (58.9% vs 41.1%), and cephalo-caudad versus lateral uterine incision extension (58 vs.39%). Earlier in 2016, a survey study from US reported that a similar number of obstetricians either "always or usually" versus "rarely or never" close parietal peritoneum (42.5% versus 46.9%, *p* = 0.46). The study also found that double-layer closure (73.3%) is most commonly used by the obstetricians to repair the uterine incision [6], which is comparable to findings of another survey study from UK which found that more than 80% of obstetricians in UK use double layer closure of the uterus while the same proportion of obstetricians utilize Pfannenstiel CS [7]. According to a recent meta-analysis of randomized controlled trials, compared to Pfannenstiel CS, Joel-Cohen–based CS is associated with reduced blood loss, operating time, time to oral intake, fever, duration of postoperative pain, analgesic injections, and time from skin incision to birth of the baby. The Joel-Cohen technique includes straight transverse incision through skin only, 3 cm below the level of the anterior superior ileac spines (higher than the Pfannenstiel incision [15]). In our study, Pfannenstiel CS is the most common technique used by the obstetricians (only 4.4% of these physicians utilized the Joel-Cohen technique), which is consistent with the findings of the UK survey study, in which more than 80% of obstetricians use this technique.

Findings of our study show that there is a wide variation in the techniques of cesarean delivery among obstetricians practicing in Ethiopia, which is consistent with previous reports from US and UK. Among the variations are closure and non-closure of the peritoneum, blunt and sharp entry, and single-layer and double-layer closure of uterine wound. The practice variation that we found is consistent with the fact that there is no evidence that shows as to which

techniques are better than the others. However, we argue that practice variation is undesirable in general because it can be confusing when different doctors/hospitals have to work together, even though the technique itself is evidence based. We recommend further analytic studies. Lacking association analysis and not being able to allocate appropriate sample size are the main limitations of our study. Not ascertaining whether every participant understood our survey questions clearly is the other limitation, though the questions seemed easy to understand. Moreover, our study lacks appropriate sample size calculation, which may negatively affect the strength of our conclusions.

In summary, our study presents the first information on the techniques of CD in the African setting. This information is helpful in boosting current evidence on this topic as and specifically in understanding how CD is practiced in Africa though it needs to be backed with further evidence from other countries within the continent. The variations in the techniques of CD found in our study is similar to the reports from high-income settings. Although available evidence (from high-income settings) doesn't' show superiority of one technique of CD over the other, there are no well-designed studies that show whether this holds true in low-income settings (such as ours). Hence, we recommend further studies which should explore the techniques of CD utilized in the rest of Africa coupled with comparative analysis of the techniques in terms of maternal and perinatal outcomes.

## Supporting information

**S1 Checklist.** *PLOS ONE* **clinical studies checklist.**
(DOCX)

## Author Contributions

**Conceptualization:** Wondimu Gudu, Zekarias Taye Sisay.

**Data curation:** Wondimu Gudu, Zekarias Taye Sisay.

**Formal analysis:** Wondimu Gudu, Mekitie Wondafrash.

**Methodology:** Wondimu Gudu, Zekarias Taye Sisay.

**Project administration:** Zekarias Taye Sisay.

**Resources:** Wondimu Gudu.

**Supervision:** Wondimu Gudu, Zekarias Taye Sisay, Abraham Fessehaye Sium.

**Validation:** Wondimu Gudu, Mekitie Wondafrash, Abraham Fessehaye Sium.

**Writing – original draft:** Zekarias Taye Sisay, Abraham Fessehaye Sium.

**Writing – review & editing:** Wondimu Gudu, Mekitie Wondafrash, Abraham Fessehaye Sium.

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
