## [Decision Letter · Decision Letter 0]

24 Jul 2023

PONE-D-23-15569Cesarean delivery surgical techniques in Africa: a survey study from EthiopiaPLOS ONE

Dear Dr. Sium,

Thank you for submitting your manuscript to PLOS ONE. After careful consideration, we feel that it has merit but does not fully meet PLOS ONE’s publication criteria as it currently stands. Therefore, we invite you to submit a revised version of the manuscript that addresses the points raised during the review process.

Please respond to all reviewers comments one by one clearly==============================

We look forward to receiving your revised manuscript.

Kind regards,

Ahmed Mohamed Maged, MD

Academic Editor

PLOS ONE

Comments from PLOS Editorial Office: We note that one or more reviewers has recommended that you cite specific previously published works. As always, we recommend that you please review and evaluate the requested works to determine whether they are relevant and should be cited. It is not a requirement to cite these works. We appreciate your attention to this request.

Reviewers' comments:

Reviewer's Responses to Questions

**Comments to the Author**

1. Is the manuscript technically sound, and do the data support the conclusions?

Reviewer #1: Yes

Reviewer #2: Yes

Reviewer #3: Yes

2. Has the statistical analysis been performed appropriately and rigorously? 

Reviewer #1: Yes

Reviewer #2: Yes

Reviewer #3: Yes

3. Have the authors made all data underlying the findings in their manuscript fully available?

Reviewer #1: Yes

Reviewer #2: Yes

Reviewer #3: Yes

4. Is the manuscript presented in an intelligible fashion and written in standard English?

Reviewer #1: Yes

Reviewer #2: Yes

Reviewer #3: No

5. Review Comments to the Author

Reviewer #1: In my opinion, the analyzed topic is interesting enough to attract the readers’ attention. I think that the abstract of this article is very clear and well structured.

In my opinion, the discussion could be studied in depth and extended. Maybe, it could be useful to evaluate the steps of hysterectomy in case of surgical complications. In particular, I suggest this article to get deeper in the topic: PMID: 36498515. Moreover, I recommend to discuss the recent pieces of evidence about the management of abdominal wall and skin closure in women undergoing primary and secondary cesarean section (authors may refer to: PMID: 29741973; PMID: 35183000).

Because of these reasons, the article should be revised and completed. Considered all these points, I think it could be of interest for the readers and, in my opinion, it deserves the priority to be published after minor revisions.

Reviewer #2: Interesting article to describe the current practice variation.

Abstract:

- Maybe add here that 99% is transverse (seems to be standard in Ethiopia, but not in all African countries, so this finding deserves a spot on the abstract)

Introduction:

- L54: I don’t think CS are safe; they are much more dangerous than vaginal birth! So maybe add: safe option when vaginal birth is dangerous to the mother or baby.

- L73 Are CS only done by obstetricians? I can’t imagine all the CS of Ethiopia are performed by these 640 gynaecologists. Maybe some are one by diploma level lower cadre healthcare workers? Or by general medical doctors? I can imagine gynaecologists are easy to recruit and perform a large part of the CS. And they teach the rest of the doctors. But if this is only a proportion of the people who perform CS you need to explain this and put as a limitation.

Methods and materials

- L 82: no need to mention the name of the report, just say according to a recent report and reference the rest.

- L89. What do you mean by all members and then randomly? A random selection or just everybody?

- L 96 Did you ask something about the number of CS performed annually?

- How did you select the topics you asked about? Did you look into the literature to find what different techniques exist? Did you ask a focus group? How do you know the techniques you asked about are all the techniques that are in use?

- Do you have any definitions about the techniques or are they all self-explanatory?

- You ask a specific question about placenta previa (what kind of incision), but it might be depending on the kind of previa (not all previa are high anterior). Why did you ask this question? Is there any literature?

- You ask about sub cutaneous dissection, but there is a difference between 1st or repeat CS. Did you take this into account?

- Did the questions have an option for an open answer in case they were not clear to the participant?

Results

- L 106. I don’t think 97% is the response rate. The response rate is 258/640. 97% of the participants filled out their questionnaire completely.

- L 124 panniculus

- L 126. What do you mean? How can you do a CS with separating the media raphe? Or you mean blunt vs sharp? And what question is this in Table 2, I don’t see it?

Discussion

- Many of the variation you find in your study are also not clear in the literature what is best to do. So is it very bad that different techniques are used? I would focus on techniques which are proven dangerous or a waste of resources (11% running locking sutures?).

- L 189 the Joel Cohen technique has proven advantages, but this is not only a different incision, but also comes with blunt dissection. I know many doctors who don’t know the exact techniques by Joel Cohen and Pfannenstiel and the differences, so I am not sure if you ask this without clarification your answers will be accurate.

- So for the discussion discuss why you asked your questions and how you made sure everyone understood what was meant. And if not: this should be mentioned as a limitation.

- What is your message with this study? You have found some variations, but many of those don’t have a clear scientific proven effect on the outcome of CS. Do you think 1 natiional technique would be better? Why? Which technique? And how would you achieve this?

But maybe for educational puproses, for young doctors trying to learn CS, one national technique would be easier. So this could be an argument. In any case it is good to say something about training. Do you know anything about the training method for CS. Are any techniques taught, or is it depending on which doctor is supervising? Is this something to improve? Maybe further studies?

Reviewer #3: I read with great interest the Manuscript titled “Cesarean delivery surgical techniques in Africa: a survey study from Ethiopia” ( PONE-D-23-15569), which falls within the aim of this Journal.

Authors should consider the following recommendations:

-Manuscript should be further revised by a native English speaker

-Inclusion/exclusion criteria should be better clarified

-The Authors did not mention the sample size calculation for their study. It is essential to specify this data in order to guarantee an adequate significance of the results obtained by the Authors.

-Was this study registered? I could not find any information about this point.

-I would suggest to add further details to discuss the maternal-fetal outcomes of vaginal birth after cesarean section (authors may refer to: PMID: 33947354; PMID: 35781586),

6. PLOS authors have the option to publish the peer review history of their article (what does this mean?). If published, this will include your full peer review and any attached files.

Reviewer #1: No

Reviewer #2: **Yes: **Rob Mooij, MD PhD

Reviewer #3: **Yes: **Doriana Lucchese

---

## [Author Response · Author response to Decision Letter 0]

24 Jul 2023

Dear reviewers,

On behalf of all authors, I would like to thank you for your extensive review along with the valuable comments, majority of which have been incorporated into the final version. Please find below listed our point-by-point responses to your comments.

Reviewer #1

In my opinion, the analyzed topic is interesting enough to attract the readers’ attention. I think that the abstract of this article is very clear and well structured.

Response: Thank you so much for appreciating our work.

In my opinion, the discussion could be studied in depth and extended. Maybe, it could be useful to evaluate the steps of hysterectomy in case of surgical complications. In particular, I suggest this article to get deeper in the topic: PMID: 36498515. Moreover, I recommend to discuss the recent pieces of evidence about the management of abdominal wall and skin closure in women undergoing primary and secondary cesarean section (authors may refer to: PMID: 29741973; PMID: 35183000).

Because of these reasons, the article should be revised and completed. Considered all these points, I think it could be of interest for the readers and, in my opinion, it deserves the priority to be published after minor revisions.

Response: Thank you very much for making this valuable comment. We have revised the manuscript accordingly. Your comment on the management of abdominal wall and skin closure in CS, is well taken and we have added statements from the references you suggested (we went one step far and referenced the original Cochrane reviews after locating them in the reference you suggested. About hysterectomy, the authors have differed adding this reference after careful consideration of your comment. As this paper focus only on the description of techniques of CS, not surgical complication of CS. We have published a separate paper on peripartum hysterectomy already (in which we described the indications and complications of it). But, thank you for your comment.

Reviewer #2

Interesting article to describe the current practice variation.

Abstract:

- Maybe add here that 99% is transverse (seems to be standard in Ethiopia, but not in all African countries, so this finding deserves a spot on the abstract)

Response: Thank you so much. You are quite right this deserves to be spotted on the abstract. As the hot points of discussion regarding the techniques of CS are the already included finding in the abstract, the authors decided to omit it. We have made a careful consideration of your comment but we thought it is better to leave it as it currently is.

Introduction:

- L54: I don’t think CS are safe; they are much more dangerous than vaginal birth! So maybe add: safe option when vaginal birth is dangerous to the mother or baby.

Response: Thank you. You are quite right CS carries far more complication than vaginal birth. We have revised the manuscript accordingly. Your comment is included as it is. 

- L73 Are CS only done by obstetricians? I can’t imagine all the CS of Ethiopia are performed by these 640 gynecologists. Maybe some are one by diploma level lower cadre healthcare workers? Or by general medical doctors? I can imagine gynecologists are easy to recruit and perform a large part of the CS. And they teach the rest of the doctors. But if this is only a proportion of the people who perform CS you need to explain this and put as a limitation.

Response: Thank you very much for flagging this important comment. You are absolutely right. We do have emergency surgical officers who perform CS in district hospitals across Ethiopia, though not in most. We agree this should be put as a limitation and we have included it in the revised version. Please note below the changes made. 

Methods and materials

- L 82: no need to mention the name of the report, just say according to a recent report and reference the rest.

Response: Thank you. We have done so.

- L89. What do you mean by all members and then randomly? A random selection or just everybody?

Response: Thank you again. We mean random selection.

- L 96 Did you ask something about the number of CS performed annually?

Response: Most of the OB-GYN included in this study practice at hospitals whose annual delivery rate is 4000 to 11,000, however we didn’t specifically collect data on accurate CS rate in the hospitals during our survey. 

- How did you select the topics you asked about? Did you look into the literature to find what different techniques exist? Did you ask a focus group? How do you know the techniques you asked about are all the techniques that are in use?

Response: We work in one of the largest hospitals in Ethiopia, which also attends one of the highest delivery 10,000-11,000 per year with a CS rate of 45-55%. Over the years of our practice we observed that there is a variation in the surgical techniques of CS among ourselves, including the most senior one(this in our morning session discussions). It has been a topic of a discussion among ourselves and we had the opportunity to make a gross observation similarity in the techniques even in the other affiliate hospitals for our medical college. We then tried to see whether there is a literature that documented the uniformity of this techniques among the other obstetricians. During our review, we discovered there was no study from Ethiopia and also from Africa specifically on this area and then we decided to form the question. 

- Do you have any definitions about the techniques or are they all self-explanatory?

Response: Thank you again. They are self-explanatory.

- You ask a specific question about placenta previa (what kind of incision), but it might be depending on the kind of previa (not all previa are high anterior). Why did you ask this question? Is there any literature?

Response: You are right it seems unclear but when we say anterior placenta previa, it is in such cases that variation in the techniques exist( some may be practicing classical cs or high low transverse or transplacental through cut and fast delivery, at least we have had simple observation that such a variation existed though not analyzed in a form of data ) otherwise in case of posterior placenta , the procedure is like in the normal ones. The is no literature but in our practice both during the old days of training and post-qualification, the approach to anterior complete low placenta previa was not uniform. 

- You ask about sub cutaneous dissection, but there is a difference between 1st or repeat CS. Did you take this into account?

Response: You are right going one step further and making such sub-group analysis would made this paper far better however we didn’t include such analysis and in our literature review, we didn’t come across to note that the data regarding the techniques of subcutaneous dissection has been stratified in such a way. Thank you for your comment.

- Did the questions have an option for an open answer in case they were not clear to the participant?

Response: Thank you so much. Yes, they had. 

Results

- L 106. I don’t think 97% is the response rate. The response rate is 258/640. 97% of the participants filled out their questionnaire completely.

Response: Thank you so much. You are quite right that this needs clarification. Out of the 640, we randomly selected 258, out these 251 had a complete response. Please check the changes made to add clarification on this in your comment above. 

- L 124 panniculus

Response: Thank you for the correction. 

- L 126. What do you mean? How can you do a CS with separating the media raphe? Or you mean blunt vs sharp? And what question is this in Table 2, I don’t see it?

Response: Thank you very much for flagging this comment. We agree it confusing as it currently stands. Yes, we mean blunt vs sharp dissection. This is an error, it doesn’t exist in Table-2. We have omitted it in the revised version.

Discussion

- Many of the variation you find in your study are also not clear in the literature what is best to do. So is it very bad that different techniques are used? I would focus on techniques which are proven dangerous or a waste of resources (11% running locking sutures?).

Response: You are quite right. Considering the scope of our paper and the need to analyze the accumulated evidence so far, we are seriously considering conducting a systematic review on techniques of CS delivery. Hence, we believe it should be a separate discussion.

- L 189 the Joel Cohen technique has proven advantages, but this is not only a different incision, but also comes with blunt dissection. I know many doctors who don’t know the exact techniques by Joel Cohen and Pfannenstiel and the differences, so I am not sure if you ask this without clarification your answers will be accurate.

Response: Thank you very much for making yet another important comment. We agree with you on this. However, with a consideration that our discussion or statements should be backed up with a reference and not go beyond the facts put in the literature, we limited our discussion to this extent only.

- So for the discussion discuss why you asked your questions and how you made sure everyone understood what was meant. And if not: this should be mentioned as a limitation.

Response: Thank you again. You comment is well taken as it and we have put it as a limitation, as you suggested. Yes, lacking a way of ascertaining whether every participant understood our questions clearly is among the main limitation.

- What is your message with this study? You have found some variations, but many of those don’t have a clear scientific proven effect on the outcome of CS. Do you think 1 national technique would be better? Why? Which technique? And how would you achieve this? But maybe for educational purposes, for young doctors trying to learn CS, one national technique would be easier. So this could be an argument. In any case it is good to say something about training. Do you know anything about the training method for CS. Are any techniques taught, or is it depending on which doctor is supervising? Is this something to improve? Maybe further studies?

Response: Thank you so much. Yes, we have found variations though many don’t have a clear scientific proven effect on the outcome of CS. Being a survey study and without a local data on the effect of this variations on CS outcomes in terms of maternal and perinatal outcomes, we don’t think our paper can put a clear recommendation on this, rather it would be good if a bigger study at national level inclusive of maternal and perinatal outcomes is conducted. Or our Ob-Gyn society or the ministry of health should assign guideline developing group and work on putting a clear practice guide though it should not lean towards standardizing to one practice technique. 

Reviewer #3

I read with great interest the Manuscript titled “Cesarean delivery surgical techniques in Africa: a survey study from Ethiopia” ( PONE-D-23-15569), which falls within the aim of this Journal.

Response: Thank you so much for your kind words of appreciation. 

Authors should consider the following recommendations:

-Manuscript should be further revised by a native English speaker

Response: Thank you. We have done so. 

-Inclusion/exclusion criteria should be better clarified

Response: We agree. We have made some changes to accommodate your comment.

-The Authors did not mention the sample size calculation for their study. It is essential to specify this data in order to guarantee an adequate significance of the results obtained by the Authors.

Response: You are quite right. Thank you so much. We believe that this should be among the main limitations of our study as we didn’t have an accurate sample size. The selection of 258 participants was a rough estimation to include Ob-Gyns at private and government obstetric centers, teaching and non-teaching hospital, and regional representation of the participants across Ethiopia, while we made sure that all participants were members of our society. We have included this as a limitation of our study in the discussion. 

-Was this study registered? I could not find any information about this point.

Response: Thank you. Being a survey study, this study was not registered instead received a local ethical clearance for a go ahead. 

-I would suggest to add further details to discuss the maternal-fetal outcomes of vaginal birth after cesarean section (authors may refer to: PMID: 33947354; PMID: 35781586)

Response: Thank you very much. It would have been good to include this but it seems to be beyond the scope of our paper, which a survey study on CS techniques variation only. Since, we didn’t analyze any longitudinal data on VBAC success rate following these techniques. Anyways, the authors appreciate your input.

---

## [Decision Letter · Decision Letter 1]

29 Aug 2023

PONE-D-23-15569R1Cesarean delivery surgical techniques in Africa: a survey study from EthiopiaPLOS ONE

Dear Dr. Sium,

Thank you for submitting your manuscript to PLOS ONE. After careful consideration, we feel that it has merit but does not fully meet PLOS ONE’s publication criteria as it currently stands. Therefore, we invite you to submit a revised version of the manuscript that addresses the points raised during the review process.

We look forward to receiving your revised manuscript.

Kind regards,

Ahmed Mohamed Maged, MD

Academic Editor

PLOS ONE

Additional Editor Comments:

Please respond to all reviewers comments one by one clearly

Reviewers' comments:

Reviewer's Responses to Questions

**Comments to the Author**

1. If the authors have adequately addressed your comments raised in a previous round of review and you feel that this manuscript is now acceptable for publication, you may indicate that here to bypass the “Comments to the Author” section, enter your conflict of interest statement in the “Confidential to Editor” section, and submit your "Accept" recommendation.

Reviewer #2: All comments have been addressed

Reviewer #3: All comments have been addressed

Reviewer #4: (No Response)

2. Is the manuscript technically sound, and do the data support the conclusions?

Reviewer #2: Yes

Reviewer #3: Partly

Reviewer #4: No

3. Has the statistical analysis been performed appropriately and rigorously? 

Reviewer #2: Yes

Reviewer #3: I Don't Know

Reviewer #4: No

4. Have the authors made all data underlying the findings in their manuscript fully available?

Reviewer #2: Yes

Reviewer #3: Yes

Reviewer #4: Yes

5. Is the manuscript presented in an intelligible fashion and written in standard English?

Reviewer #2: Yes

Reviewer #3: No

Reviewer #4: Yes

6. Review Comments to the Author

Reviewer #2: Thank you for the revised version of the manuscript. You have addressed the concerns and the manuscript has improved.

Here are just some small suggestions:

Abstract:

L 46: I am not sure of this English is correct. I would say: “Blunt and sharp, cc and lateral, closure and non-closure… “

Introduction

L. 71 I don’t like the term developed world, it is a bit condescending. I would just say High Income Countries.

L 74 I don’t know what you mean with Among the SSA and Ethiopia representing something. Maybe just remove it.

Discussion

L 172, recruited, found

In the second paragraph you say there is no difference and in the conclusions you iterate that these options are evidence based. This renders your research a bit pointless. And is the weakest point of the manuscript in my opinion. Why are you researching something that doesn’t matter? Also in the introduction this is not very clear, you just say it is unknown, not why is should be known. Maybe you can argue that practice variation is undesirable in general because it can be confusing when different doctors/hospitals have to work together, even though the technique itself is evidence based. Or for training-purposes. Or at least you can say that the practice variation that you found is consistent with the fact that there is no evidence which technique is better. If all doctors would experience one technique is better you would end practice variation.

Reviewer #3: I read with great interest the Manuscript titled “Cesarean delivery surgical techniques in Africa: a survey study from Ethiopia

” (PONE-D-23-15569R1), which falls within the aim of this Journal.

Authors should consider the following recommendations:

-Manuscript should be further revised by a native English speaker.

-Inclusion/exclusion criteria should be better clarified.

-What are the actual clinical implications of this study? it is important to report the results obtained by the authors in the context of clinical practice and to adequately highlight what contribution this study adds to the literature already existing on the topic and to future study perspectives.

Reviewer #4: Please clearly explaine how contribution the results of your study can provide in the literature

LİNE 83-86 İN Mat&met the information about Ethşopia should be placed in introduction

Please the advantages of techniques explain clearly in introduction

Please clearly explain which procedures related CS were evaluated during the survey questionnaire such as antiseptics for the preparation of skin, the prophylactic antibiotics choices

There is no statistical analysis information in mat&met

Please explain the advantages and desadvantages of each techniques in discussion when you presented the results of literatures related to each techniques

7. PLOS authors have the option to publish the peer review history of their article (what does this mean?). If published, this will include your full peer review and any attached files.

Reviewer #2: **Yes: **Rob Mooij

Reviewer #3: **Yes: **Dott.ssa Doriana Lucchese

Reviewer #4: No

---

## [Author Response · Author response to Decision Letter 1]

31 Aug 2023

Dear Reviewers,

I would like to thank you very much on behalf of all authors for your extensive review along with the valuable comments, majority of which are well taken as they are and included in the revised version. Please find below listed our point-by-point response to all your comments. 

Reviewer #2: 

Thank you for the revised version of the manuscript. You have addressed the concerns and the manuscript has improved.

Here are just some small suggestions:

Abstract:

L 46: I am not sure of this English is correct. I would say: “Blunt and sharp, cc and lateral, closure and non-closure… “

Response: Thank you very much for making this valuable comment. It has been corrected accordingly. 

Introduction

L. 71 I don’t like the term developed world, it is a bit condescending. I would just say High Income Countries.

Response: Thank you. It has been corrected accordingly. 

L 74 I don’t know what you mean with Among the SSA and Ethiopia representing something. Maybe just remove it.

Response: You are quite right. We mean that Ethiopia as the second largest population in the Sub-Saharan region with high fertility rate, it represents the region with this large burden of obstetric patients and procedures, including CS. We have modified the English a little bit but leaving it as it may be okay. 

Discussion

L 172, recruited, found

Response: Thank you for noting this silly mistake. I have separated the statement with a colon.

In the second paragraph you say there is no difference and in the conclusions you iterate that these options are evidence based. This renders your research a bit pointless. And is the weakest point of the manuscript in my opinion. Why are you researching something that doesn’t matter? Also in the introduction this is not very clear, you just say it is unknown, not why is should be known. Maybe you can argue that practice variation is undesirable in general because it can be confusing when different doctors/hospitals have to work together, even though the technique itself is evidence based. Or for training-purposes. Or at least you can say that the practice variation that you found is consistent with the fact that there is no evidence which technique is better. If all doctors would experience one technique is better you would end practice variation.

Response: Thank you very much for this very constructive comment. I have taken your argument with slight modification and replaced the previous argument. You are right, “Why need our study then?”. I have also modified the statement at the introduction section as you suggested. 

Reviewer #3: 

I read with great interest the Manuscript titled “Cesarean delivery surgical techniques in Africa: a survey study from Ethiopia

” (PONE-D-23-15569R1), which falls within the aim of this Journal.

Response: Thank you so much for appreciating our work.

Authors should consider the following recommendations:

-Manuscript should be further revised by a native English speaker.

Response: Thank you. We have do so accordingly.

-Inclusion/exclusion criteria should be better clarified.

Response: Thank you so much for your comment. I have modified it accordingly. 

-What are the actual clinical implications of this study? it is important to report the results obtained by the authors in the context of clinical practice and to adequately highlight what contribution this study adds to the literature already existing on the topic and to future study perspectives.

Response: Thank you very much for your valuable comment. Apologies, a complete one paragraph was missed from the end in the discussion section. I have included your comment in the revised version. Please check the last paragraph in the discussion section.

Reviewer #4: 

Please clearly explain how contribution the results of your study can provide in the literature

Response: Thank you very much for raising this important comment. As it was also raised by the other reviewer. I have addressed it well in the revised version. Please check the last paragraph in the discussion section.

LİNE 83-86 İN Mat&met the information about Ethiopia should be placed in introduction

Response: Thank you so much. I have moved the statement regarding maternal mortality in Ethiopia to introduction but left the information about CS rate and the Ethiopian society of obstetricians and gynecologists in the methods section. One, there is enough information regarding CS in the country in the introduction and the information regarding ESOG we thought it fits well in the methods, as it is the source of our data.

Please the advantages of techniques explain clearly in introduction

Response: As our study focused on describing the variation in the techniques without analysis of procedure outcomes (advantages and disadvantages) we didn’t discuss about it in the introduction. However, we have mentioned about it based on available evidence in the discussion, where it fits more, as we need to have a say regarding the implication of the findings. 

Please clearly explain which procedures related CS were evaluated during the survey questionnaire such as antiseptics for the preparation of skin, the prophylactic antibiotics choices

Response: Thank you so much for flagging this important comment. I have added as a statement in bracket in the methods and materials section. Please check in the third paragraph. 

There is no statistical analysis information in mat&met

Response: Thank you. The statistical analysis, which a simple descriptive analysis on SPSS version 22 is well described in the methods section as well as in the abstract section. 

Please explain the advantages and disadvantages of each techniques in discussion when you presented the results of literatures related to each techniques

Response: Thank you so much once again for another important comment. We have included the available evidence on this topic in the discussion technique. Please check paragraph 2 and 4 in the discussion section. We couldn’t provide further information as these are the only available studies that compared outcomes of the techniques.

---

## [Decision Letter · Decision Letter 2]

14 Sep 2023

PONE-D-23-15569R2Cesarean delivery surgical techniques in Africa: a survey study from EthiopiaPLOS ONE

Dear Dr. Sium,

Thank you for submitting your manuscript to PLOS ONE. After careful consideration, we feel that it has merit but does not fully meet PLOS ONE’s publication criteria as it currently stands. Therefore, we invite you to submit a revised version of the manuscript that addresses the points raised during the review process.

please respond to all reviewer comments one by one==============================

We look forward to receiving your revised manuscript.

Kind regards,

Ahmed Mohamed Maged, MD

Academic Editor

PLOS ONE

Reviewers' comments:

Reviewer's Responses to Questions

**Comments to the Author**

1. If the authors have adequately addressed your comments raised in a previous round of review and you feel that this manuscript is now acceptable for publication, you may indicate that here to bypass the “Comments to the Author” section, enter your conflict of interest statement in the “Confidential to Editor” section, and submit your "Accept" recommendation.

Reviewer #3: All comments have been addressed

Reviewer #4: All comments have been addressed

2. Is the manuscript technically sound, and do the data support the conclusions?

Reviewer #3: Yes

Reviewer #4: Yes

3. Has the statistical analysis been performed appropriately and rigorously? 

Reviewer #3: Yes

Reviewer #4: Yes

4. Have the authors made all data underlying the findings in their manuscript fully available?

Reviewer #3: Yes

Reviewer #4: Yes

5. Is the manuscript presented in an intelligible fashion and written in standard English?

Reviewer #3: (No Response)

Reviewer #4: Yes

6. Review Comments to the Author

Reviewer #3: Authors should consider the following recommendations:

-Inclusion/exclusion criteria should be better clarified.

-Manuscript should be further revised by a native English speaker

Reviewer #4: there is no additional comment. the authors have adequately addressed mycomments. it can be accepted with revison version

7. PLOS authors have the option to publish the peer review history of their article (what does this mean?). If published, this will include your full peer review and any attached files.

Reviewer #3: No

Reviewer #4: No

---

## [Author Response · Author response to Decision Letter 2]

15 Sep 2023

Dear Reviewers,

Thank you very much again for your valuable comments. Please find below listed the authors response your comments.

Reviewer #3

-Inclusion/exclusion criteria should be better clarified.

Response: Thank you very much. We have modified the inclusion and exclusion criteria accordingly.

-Manuscript should be further revised by a native English speaker

Response: Thank you very much. The manuscript has been edited extensively by a native English speaker.

Reviewer #4

There is no additional comment. the authors have adequately addressed mycomments. it can be accepted with revison version

Response: Thank you very much!

---

## [Editor Report · Decision Letter 3]

19 Sep 2023

Cesarean delivery surgical techniques in Africa: a survey study from Ethiopia

PONE-D-23-15569R3

Dear Dr. Sium,

We’re pleased to inform you that your manuscript has been judged scientifically suitable for publication and will be formally accepted for publication once it meets all outstanding technical requirements.

Kind regards,

Ahmed Mohamed Maged, MD

Academic Editor

PLOS ONE
---

## [Editor Report · Acceptance letter]

28 Sep 2023

PONE-D-23-15569R3 

Cesarean delivery surgical techniques in Africa: a survey study from Ethiopia 

Dear Dr. Sium:

I'm pleased to inform you that your manuscript has been deemed suitable for publication in PLOS ONE. Congratulations! Your manuscript is now with our production department. 

Kind regards, 

on behalf of

Professor Ahmed Mohamed Maged 

Academic Editor

PLOS ONE